# Singular Value Fine-tuning for Few-Shot Class-Incremental Learning

## Abstract

Class-Incremental Learning (CIL) aims to learn knowledge from new classes sequentially, while rataining the knowledge obtained from previously encountered classes, thereby mitigating the challenge of Catastrophic Forgetting. In a more realistic scenario, future unseen classes may contain only a few samples, leading to a new challenge of over-fitting, which is referred to as Few-Shot Class-Incremental Learning (FSCIL). Existing works explore FSCIL from various perspectives, such as classifier calibration and backbone extension. Most of them treat the many-shot base session and incremental few-shot sessions separately, as the model tends to overfit on few-shot classes. In this paper, we propose Singular Value Fine-tuning for few-shot Class-incremental Learning (SVFCL) to constantly learn base and incremental sessions based on the pre-trained ViT encoder. SVFCL incorporates incremental adapters, each of which is attached to a corresponding pre-trained module and contains only a small number of learnable parameters, effectively reducing the risk of overfitting. Furthermore, since each adapter is task-specific, information from previous tasks is well-preserved, mitigating catastrophic forgetting. Our experimental results demonstrate that SVFCL achieves substantial improvements over state-of-the-art methods while requiring significantly less computational overhead and epochs.

## 1 Introduction

Given the unprecedented increase in computational resources and data availability, the ability to continuously learn from new data while retaining previously acquired knowledge is crucial for developing truly intelligent systems. Class-Incremental Learning (CIL) (Zhou et al., 2023; Rebuffi et al., 2017; Li & Hoiem, 2017; Kirkpatrick et al., 2017) seeks to address this challenge by introducing new classes sequentially, requiring the model to adapt to changes in data distribution while preserving knowledge of previously learned classes. However, the practical implementation of CIL is limited by the difficulty of acquiring sufficiently large datasets for new classes, which significantly constrains its application in real-world scenarios.

Few-Shot Class-Incremental Learning (FSCIL) (Tao et al., 2020; Tian et al., 2024b; Zhang et al., 2021; Peng et al., 2022; Zhou et al., 2022; Yang et al., 2023) focuses on the incremental learning scenario where newly introduced classes have only a limited number of samples. Unlike traditional CIL, FSCIL begins by training on a task with a sufficiently large dataset to establish prior knowledge, followed by subsequent tasks that include only a small number of samples. This scenario highlights the model's ability to sequentially learn new classes from limited samples while retaining previously learned knowledge, thereby enhancing its practical applicability in real-world contexts. Conversely, the practical setting of learning incrementally with limited samples also introduces two key problems in the demanding FSCIL scenario:

- ○ (Forgetting) During the sequential training, the model will experience a significant drop in performance on previously learned tasks after training on new tasks.
- ○ (Overfitting) Due to the limited data of the new tasks, the model may overfit to these examples, which negatively impacts its performance on these tasks.

Most previous studies primarily employed shallow models such as ResNet-18, operating solely with a constrained set of training parameters to address these challenges. Specifically, Zhang et al. (2021) first proposed decoupling the training phases of representations and classification head to effectively

mitigate forgetting. Subsequently, Shi et al. (2021) highlighted that pre-training on the base session is more critical than adapting to new sessions, which inspired a wide range of works that focus on training the backbone exclusively on the base session while tuning the classification head on the subsequent few-shot sessions (Zhou et al., 2022; Peng et al., 2022; Wang et al., 2024; Ahmed et al., 2024; Yang et al., 2023; Akyürek et al., 2021; Hersche et al., 2022). However, these methods based on shallow models are inadequate for effectively transferring domain knowledge from the base session to the few-shot sessions.

Recently, many studies have applied large pre-trained models, such as Vision Transformer (ViT) (Alexey, 2020), to FSCIL, as these models can provide a backbone with strong generalization capabilities (Liu et al., 2024; Park et al., 2024; D'Alessandro et al., 2023). Most of these works adopt a prompt learning paradigm, which involves designing specific input prompts to guide the pre-trained backbone to perform well across various few-shot sessions. However, these approaches typically utilize a completely frozen backbone, which renders performance highly dependent on the underlying model's capabilities and, consequently, may result in inferior performance when suboptimal models are employed. On the flip side, attempting to fine-tune all parameters of these large models can lead to catastrophic overfitting, especially considering the extremely limited samples available in few-shot sessions. This arises a critical question: *how can we effectively fine-tune large models to address the challenging issue of overfitting in FSCIL?*

To address this challenge, this paper proposes a simple yet effective framework, called Singular Value Fine-tuning for Class-incremental Learning (SVFCL), to simultaneously mitigate catastrophic forgetting and overfitting. The core idea of this method is to treat both the many-shot base session and the subsequent few-shot sessions equally by adding task-specific adapters for each session. This strategy facilitates the retention of prior knowledge within these adapters, effectively overcoming forgetting by enabling the model to access previously learned information during the continual learning stage. To prevent the adapters from introducing a substantial parameter load that could lead to overfitting in few-shot sessions, we further propose a singular value fine-tuning strategy, which decomposes the pre-trained weights—such as those of the linear layers in ViT—using Singular Value Decomposition (SVD) to obtain two singular matrices and singular values, and tunes parameters solely in the singular value positions for each incremental session. This approach reduces the number of learnable parameters, thereby minimizing the risk of overfitting; furthermore, it provides valuable insights into how parameters in the inner layers of ViT evolve during incremental training. Overall, the main contributions of this paper are summarized as follows:

- We propose the SVFCL, a simple yet effective strategy to jointly mitigate both catastrophic forgetting and overfitting in the FSCIL scenario.
- We design lightweight adapters based on singular value fine-tuning to capture task-specific knowledge for each incremental session, which introduces quite few learnable parameters and thereby alleviating overfitting in FSCIL.
- We conduct extensive experiments on three commonly used datasets, where the proposed SVFCL achieves new state-of-the-art results in FSCIL, demonstrating its effectiveness and robustness.

## 2 RELATED WORKS

**Continual learning** aims to address the catastrophic forgetting problem (Zhou et al., 2023). Conventional class-incremental learning methods can be divided into four mainstream categories as follows. First, regularization-based methods (Kirkpatrick et al., 2017; Chaudhry et al., 2018; Zenke et al., 2017; Aljundi et al., 2018) assume that network parameters have varying importance for different tasks. The essence of these methods is to allocate importance to parameters and impose penalties in loss items to preserve old knowledge. Second, replay-based methods (Rebuffi et al., 2017; Buzzega et al., 2020; De Lange & Tuytelaars, 2021; Chaudhry et al., 2018; Bang et al., 2021) mitigate catastrophic forgetting by maintaining some old exemplars in the memory buffer during current training. However, these methods often fall short due to real-world constraints, such as data privacy and storage limitations. Third, while Hinton et al. (2015) proposed Knowledge Distillation (KD) technology to transfer knowledge from a teacher model to a student model, some methods (Li & Hoiem, 2017; Lee et al., 2019; Hou et al., 2018; Zhai et al., 2019; Dhar et al., 2019) incorporate knowledge distillation to constrain model update. They view the old model as a teacher and the

current model as a student and alleviate catastrophic forgetting by transferring knowledge from the old model to the current model. Fourth, architecture-based methods (Yan et al., 2021; Li et al., 2019; Rusu et al., 2016) typically expand the network structure to enhance its capacity for learning specific tasks.

**Parameter-Efficient-Fine-Tuning (PEFT)** aims to adapt large pre-trained models to diverse downstream tasks efficiently by incorporating extra lightweight learnable parameters while freezing the pre-trained backbone, thereby circumventing the computational and storage costs associated with direct fine-tuning of large models. With the rapid development of large pre-trained models, a growing number of PEFT methods have emerged, such as prefix tuning (Li & Liang, 2021), prompt tuning (Lester et al., 2021; Jia et al., 2022), adapter tuning (Houlsby et al., 2019), scale and shift tuning (SSF) (Lian et al., 2022), low-rank tuning (Hu et al., 2021; Sun et al., 2022), and so on.

In the field of class-incremental learning, approaches incorporating PEFT are becoming increasingly popular. L2P (Wang et al., 2022b) first proposes the prompt pool to select task-specific prompts during incremental learning. DualPrompt (Wang et al., 2022a) further introduces task-invariant prompts to learn knowledge shared between diverse tasks. Numerous subsequent prompt-based continual learning approaches (Liu et al., 2024; D'Alessandro et al., 2023; Tian et al., 2024a) have adopted the aforementioned strategy of combining task-invariant and task-specific prompts. CodaP Smith et al. (2023) further enhances the training and selecting strategies in an end-to-end manner.

**Few-Shot Class-Incremental Learning (FSCIL)** was first proposed by Tao et al. (2020) and is expected to incrementally learn new few-shot tasks while given at first a base task with abundant samples for base training. Panos et al. (2023) also explores a more extreme scenario in which no base task with sufficient data is given and only a small number of data are accessible in all sessions, including the first. Following Tian et al. (2024b), existing FSCIL methods can be roughly categorized into five groups: traditional machine learning methods, meta-learning-based methods, feature-and-feature-space-based methods, replay-based methods and architecture-based methods. Additionally, another taxonomy divides FSCIL methods into two categories: base-focus (Zhang et al., 2021; Zhou et al., 2022; Wang et al., 2024; Yang et al., 2023; Akyürek et al., 2021; Hersche et al., 2022; Ahmed et al., 2024; Peng et al., 2022) and incremental-focus (Kim et al., 2024; Tao et al., 2020), in terms of their training emphasis. This distinction is supported by empirical evidence from studies such as Zhang et al. (2021) and Shi et al. (2021), which demonstrate that a backbone trained on the base session is more critical than merely calibrating the feature space for updating knowledge in incremental sessions. However, these methods mainly rely on shallow models, which are insufficient to effectively transfer domain knowledge from the base session. This limitation prompts our focus on fine-tuning foundation models in this paper.

**FSCIL methods based on foundation models** have gradually attracted attention in recently years. Leveraging the generalization ability of foundation models, these methods can quickly adapt to incremental tasks and achieve remarkable performance. We can roughly categorize these methods in terms of their backbones, i.e. Vision Transformer (ViT) (Alexey, 2020) or large Vision-Language-Model (VLM) such as CLIP (Radford et al., 2021). Liu et al. (2024) propose the Attention-aware Self-adaptive Prompt (ASP) framework with ViT as its network backbone. ASP utilizes task-invariant prompts to capture shared knowledge and self-adaptive task-specific prompts to obtain specific information. However, ASP suffers from a high training cost, as it requires an extra pass over the training data in each epoch to cluster class centers. Compared to ASP, our proposed SVFCL introduces a lightweight computational overhead and benefits from enhanced training efficiency. There are also works that employ vision and language models. Specifically, Park et al. (2024) propose PriViLege, which combines prompting functions and knowledge distillation. D'Alessandro et al. (2023) propose CPE-CLIP, a CLIP-based framework that utilizes prompting techniques to generalize pre-trained knowledge to incremental sessions. Different from these methods, our approach does not incorporate any textual priors yet achieves competitive performance.

## 3 PRELIMINARY

### 3.1 PROBLEM FORMULATION

Suppose there is a sequence of downstream tasks $\{\mathcal{D}^0, \mathcal{D}^1, \cdots, \mathcal{D}^T\}$, where the $t$-th task $\mathcal{D}^t = \{(x_i^t, y_i^t)\}_{i=1}^{N_t}$ consists of $N_t$ samples $x_i^t$ and corresponding labels $y_i^t \in C^t$. Following the disjoint

FSCIL settings in Tao et al. (2020), we adopt the non-overlapping label spaces $C^t$ for different tasks, i.e., $C^i \cap C^j = \emptyset$ for all $i \neq j$, where $i, j \in \{0, 1, ..., T\}$.

Different from the CIL, FSCIL aims to incrementally learn new classes with only a few samples per class. Typically, FSCIL begins with a base session $\mathcal{D}^0$ containing sufficient data, while the subsequent few-shot sessions $\mathcal{D}^t(t > 0)$ include only a few training samples for new classes in $C^t$. For instance, in an $N$-way $K$-shot FSCIL scenario, each incremental session $\mathcal{D}^t(t > 0)$ consists of $N$ classes, with each class containing $K$ samples. The objective of FSCIL is to ensure the model performs well on both the fully-sampled base session and the subsequent few-shot sessions with limited examples. This can be formally expressed as

$$\min \mathbb{E}_{(\mathbf{x},y)\in\mathcal{D}^0\cup\mathcal{D}^1\cup\cdots\cup\mathcal{D}^T}\mathcal{L}(f(\mathbf{x}), y) = \min_{\theta,w} \mathbb{E}_{(\mathbf{x},y)\in\mathcal{D}^0\cup\mathcal{D}^1\cup\cdots\cup\mathcal{D}^T}\mathcal{L}(g_w \circ \phi_\theta(\mathbf{x}), y), \quad (1)$$

where $\mathcal{L}(\cdot, \cdot)$ denotes the loss function that measures the discrepancy between prediction and ground-truth label. The model $f$ can be further decomposed into an image encoder $\phi_\theta$ parameterized by $\theta$ and a classifier $g$ parameterized by $w$, such that $f = g_w \circ \phi_\theta$.

## 3.2 SINGULAR VALUE DECOMPOSITION

Singular Value Decomposition (SVD) is a mathematical technique used for the factorization of matrices, which provide a powerful tool for revealing intrinsic properties and structures within datasets. Specifically, SVD states that any matrix $M \in \mathbb{R}^{m \times n}$ can be expressed as:

$$M = USV^\top, \quad (2)$$

where $U$ is an $m \times m$ orthogonal matrix whose columns are the left singular vectors of $M$; $S$ is an $m \times n$ diagonal matrix containing the non-negative singular values of $M$; $V^\top$ is the transpose of an $n \times n$ orthogonal matrix whose columns are the right singular vectors of $M$. Note that the number of singular values is equal to the rank of $M$, which play an important role in providing information about the intrinsic dimensionality of $M$.

Assuming the rank of $M$ is $r$, it can also be reconstructed as a sum of rank-one matrices, i.e., $M = \sum_{i=1}^{r} s_i\mathbf{u}_i\mathbf{v}_i^\top$, where $s_i$ is the $i$-th singular value of $M$, and $u_i$ and $v_i$ are its corresponding left and right singular vectors, respectively.

## 3.3 NEAREST CLASS MEAN (NCM) CLASSIFIER

Snell et al. (2017) propose prototypical networks for the challenge of few-shot classification, which has been proven to be beneficial to improve classification performance in few-shot class-incremental learning (Zhou et al., 2022; Wang et al., 2024). The NCM classifier predicts the label for a sample $\mathbf{x}$ in terms of the similarities between its embedding and prototypes of all seen classes, in the form:

$$y^* = \arg\min_{i\in\mathcal{C}} d(\phi_\theta(\mathbf{x}), \mathbf{p}_i), \quad (3)$$

where $\mathcal{C}$ indicates the set of all seen classes, $d(\cdot, \cdot)$ is a distance function, $\phi_\theta$ is the backbone and $\mathbf{p}_i, i \in \mathcal{C}$ are prototypes computed by the class mean of embeddings:

$$\mathbf{p}_c = \frac{1}{N_c} \sum_{i=1}^{N_c} \phi_\theta(\mathbf{x})\mathbb{1}\{y = c\}, \quad (4)$$

in which $\mathbf{p}_c$ is the prototype of the class $c$, $N_c$ is the number of the class $c$, and $\mathbb{1}\{\cdot\}$ is the identity function. In detail, we utilize the cosine classifier during training and replace the classifier weights with prototypes of all seen classes at the end of a session.

## 4 METHODS

As aforementioned, FSCIL methods are expected to obtain a well-generalized base model trained at the base session. Intuitively, the pre-trained foundation models are a good choice to provide us with prior knowledge to transfer to the following incremental few-shot sessions. Thus, our method focuses on parameter-efficient fine-tuning to utilize the pre-trained models. Although there have been

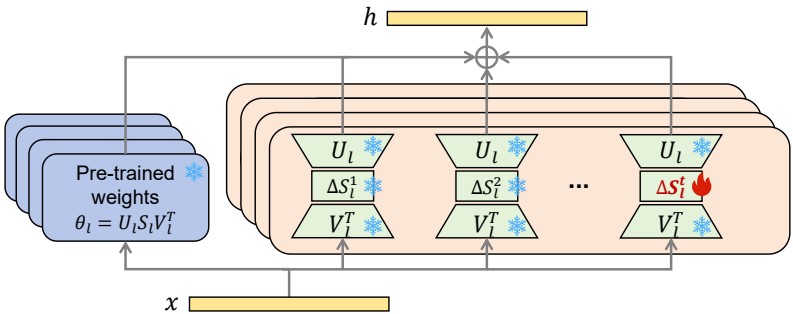

Figure 1: Visualization comparison of incremental attention maps generated by Grad-CAM (Selvaraju et al., 2017) between Full-tuning (Top) and our method (Bottom) on the miniImageNet dataset. Our approach directs attention towards crucial features, whereas the Full-tuning method struggles to learn effective representations. More visualization results are presented in Appendix A.

Figure 2: Illustration of our proposed SVFCL framework during the $t$-th incremental session. The left module in light blue denotes partial weights of the pre-trained backbone, which are required to be tuned in subsequent incremental sessions. For each parameter matrix $\theta_l$, its singular value decomposition is $\theta_l = U_l S_l V_l^\top$. The right modules are incremental adapters. For the new session $t$, we introduce a new learnable adapter $U_l \Delta S_l^t V_l^\top$ of the same size as $\theta_l$, and we only fine-tune the singular values, i.e., $\Delta S_l^t$, while keeping the other parameters $U_l$ and $V_l^\top$ frozen. During the forward pass of each module, we compute the summation after the input passes through the original pre-trained weights and all previously seen adapters.

numerous attempts to leverage pre-trained foundation models in the context of continual learning, the exploration of the foundation model in FSCIL remains limited, and the majority of these attempts employ prompt-tuning techniques (Liu et al., 2024; Park et al., 2024; D'Alessandro et al., 2023). These approaches commonly utilize a completely frozen backbone, which limits the representational ability of the model and renders its performance highly dependent on the underlying foundation model's capabilities.

Unlike previous prompt-tuning-based methods, we resort to fine-tuning the backbone of the foundation model and fully leveraging its adaptability for incremental tasks during the FSCIL process. A natural solution is to fine-tune all parameters of the backbone; however, this approach may lead to unsatisfactory performance in few-shot sessions due to the extremely limited number of available samples. We attribute this phenomenon to the issue of catastrophic overfitting. As illustrated in Fig. 1, the Full-tuning method struggles to learn effective features and tends to overfit to local areas. In contrast, our method learns critical features during the FSCIL process.

To address the challenge of overfitting, this paper aims to propose Singular Value Fine-tuning for Class-incremental Learning (SVFCL), a simple yet effective framework to efficiently mitigate the overfitting issue. Our main idea is to introduce a lightweight adapter for each parameter matrix during each incremental session, wherein the adapter is designed to tune only the singular values of the parameter matrix. Specifically, for a parameter matrix $\theta$, we first compute its singular value decomposition as $\theta = USV^\top$ before the FSCIL process. Then, for the $t$-th incremental session, the singular value fine-tuning of $\theta$ is defined as

$$\theta^t = \theta^{t-1} + \Delta\theta = \theta^{t-1} + U\Delta S^t V^\top, \quad (5)$$

where $\Delta S^t$ is a diagonal matrix and only its diagonal entries are learnable parameters, and we define $\theta^0 = \theta$ when $t = 1$. In other words, during the $t$-th incremental session, we only tuning the singular

---

**Algorithm 1** Training Algorithm of the Proposed SVFCL

---

1: **Input:** A sequence of tasks $\{\mathcal{D}^0, \mathcal{D}^1 \cdots, \mathcal{D}^T\}$, pre-trained backbone $\phi_\theta$, random initialized cosine classifier $g_\omega$, candidate set $\Theta = \cup_{l \in \mathcal{J}} \theta_l$ for tuning.
  # Before FSCIL
2: **for** $l \in \mathcal{J}$ **do**
3:   Compute the singular value decomposition of $\theta_l$: $\theta_l = U_l S_l V_l^\top$
4: **end for**
  # During FSCIL
5: **for** each task **do**
6:   Add a new adapter for each candidate: $\theta_l \leftarrow \theta_l + U_l \Delta S_l V_l^\top$.
7:   Freeze all parameters except $\Delta S_l$ and $\omega$.
8:   Do forward process by Eq. 6.
9:   Update $\Delta S_l$ and $\omega$ by Eq. 1
10:   **for** each new class $c$ **do**
11:    Calculate prototype $\mathbf{p}_c$ by Eq. 4
12:    Update classifier: $\omega[c] \leftarrow \mathbf{p}_c$
13:   **end for**
14: **end for**
15: **return** $\theta, \omega$

---

values of the original parameter matrix $\theta$. We illustrate our SVFCL in Fig. 2, and we can see that both $\theta^{t-1}$ and $\Delta\theta$ are multiplied with the same input, and their respective output vectors are element-wisely summed. For $h = \theta^t x$, our modified forward pass yields:

$$
\begin{aligned}
h = \theta^{t-1}x + \Delta\theta x &= \theta^{t-1}x + U\Delta S^t V^\top x \\
&= \theta^{t-2}x + U\Delta S^{t-1}V^\top x + U\Delta S^t V^\top x \\
&= U(S + \Delta S^1 + \cdots + \Delta S^t)V^\top x.
\end{aligned}
\tag{6}
$$

According to Eq. 5 and 6, we can conclude that the proposed SVFCL framework has the following advantages:

- *Efficient Reduction of Overfitting.* During each incremental session, we only tune the singular values of the original parameter matrix $\theta$. The number of these parameters is quite few compared to that of the original matrix, which greatly reduces the risk of the model overfitting on few-shot session tasks. Additionally, we can further reduce the number of parameters by applying low-rank approximation in SVD. [1]

- *Forgetting Alleviation for Old Tasks.* We refrain from updating the singular matrices $U$ and $V$, indicating that we maintain the same singular vectors (semantic clues) across all incremental sessions. This strategy effectively alleviates the risk of feature shift, thereby mitigating the phenomenon of forgetting.

- *No Additional Inference Latency.* During testing, we can explicitly compute and store $\theta^t = \theta^{t-1} + U\Delta S^t V^\top$ and perform inference as usual. This guarantees that we do not introduce any additional latency during inference.

The training algorithm of our SVFCL is listed in Algorithm 1, and we list the architectural implementation details as follows.

**Architectural Details.** In detail, let the set of parameter matrices that be denoted as $\Theta$, which can be partitioned into a collection of disjoint subsets, expressed as $\Theta = \bigcup_{l \in \mathcal{J}} \theta_l$, where $\mathcal{J}$ is the index set. Each parameter matrix corresponds to a specific block or module within the backbone, such as the query, key, and value matrices in Multi-Head Self-Attention (MSA) blocks, fully connected linear layers in Multi-Layer Perceptron (MLP) modules, and other similar components. Zhang et al. (2023) pointed out that fine-tuning the Feed-Forward Network (FFN), i.e. a two-layer MLP, is

---

[1]The low-rank SVD only retains the first few singular values and their corresponding singular vectors, which can greatly reduce the number of learnable parameters. LoRA (Hu et al., 2021) is a widely-used low-rank fine-tuning technique, while with the same rank, our number of learnable parameters can always be smaller than that of LoRA. We explore the efficiency of our low-rank fine-tuning in Fig. 4.

more effective than the self-attention module. Following Zhang et al. (2023), we merely fine-tune parameters in MLP, more specifically, the two fully-connected linear layers. In this paper, we use the ViT-B/16-1K (Alexey, 2020) as our backbone, which contains 12 blocks, each of which includes a MLP module. Suppose $\{1, 2, 3, \cdots, 12\}$ is the index set of MLP modules. We just apply SVFCL to a subset $\mathcal{J}'$ of $\{1, 2, 3, \cdots, 12\}$. $\theta_l$ belongs to either of the two fully-connected layers whithin the corresponding MLP module and $\mathrm{SVD}(\theta_l) = U_l S_l V_l^{\top}$ as shown in Fig. 2.

## 5 EXPERIMENTS

### 5.1 EXPERIMENTAL SETTINGS

**Datasets.** We conduct experiments on three widely used datasets: CIFAR100 (Krizhevsky et al., 2009), CUB200-2011 (Wah et al., 2011) and miniImageNet (Ravi & Larochelle, 2016), which are detailed as follows:

- **CIFAR100** is a labeled subset of the 80 million tiny images dataset, consisting of 60000 $32\times32$ colored images in 100 classes with 600 images each. There are 500 training images and 100 testing images per class. We split CIFAR100 into 9 sessions where the first base session contains 60 classes with all training samples and the remaining 8 few-shot sessions are 5-way 5-shot, i.e. each new session contains 5 classes and 5 samples per class.
- **Caltech-UCSD Birds-200-2011 (CUB200-2011)** is a fine-grained bird dataset, containing 11788 colored images in 200 classes. There are 5994 training images and 5794 testing images. We use 100 classes with all training images as the base session. The remaining 100 classes are divided into 10 incremental new sessions with 10-way 5-shot each, i.e. each new session contains 10 classes and 5 samples per class.
- **miniImageNet** is a subset of the ImageNet dataset, consisting of 60000 colored images in 100 classes with 600 images each. We use 60 classes as the base session and the remaining 40 classes are split into 8 incremental new sessions with 5-way 5-shot each, i.e. each new session contains 5 classes and 5 samples per class.

**Evaluation Protocol.** Following D'Alessandro et al. (2023), we assess the performance using two evaluation metrics, including Average Accuracy ($A_{avg}$) and Performance Dropping rate ($PD$). Given the Top-1 accuracy for the base task $A_0$ and that for the $t$-th few-shot task $A_t$, the two metrics are defined as follows: (1) $A_{avg}$, computed as $A_{avg} = \frac{1}{T} \sum_{t=1}^{T} A_t$, which measures the mean performance across all the seen tasks after training the $T$-th task; and (2) $PD$, defined as $PD = A_0 - A_T$, which quantifies the extent of forgetting after training the $T$-th task.

**Baselines.** We mainly compare our proposed method with recent state-of-the-art approaches, which can be broadly categorized into four groups: (1) Conventional continual learning method: iCaRL (Rebuffi et al., 2017); (2) Continual learning methods based on foundation models: L2P (Wang et al., 2022b), DualPrompt (Wang et al., 2022a), CodaP (Smith et al., 2023); (3) Conventional FSCIL methods: CEC (Zhang et al., 2021), FACT (Zhou et al., 2022), TEEN (Wang et al., 2024); (4) FSCIL methods based on foundation models: ASP (Liu et al., 2024), CPE-CLIP (D'Alessandro et al., 2023), PriViLege (Park et al., 2024).

**Implementation Details.** Our experiments are all implemented by Pytorch (Paszke et al., 2019) on NVIDIA GeForce RTX 4090. We choose ViT-B/16-1K (Alexey, 2020) as the pre-trained backbone. As aforementioned, we replace the original linear classifier with the NCM classifier after the pre-trained model. Additionally, the input images of each dataset are resized to $224\times224$. To alleviate overfitting in FSCIL, we train the base session task for only 3 epochs and each few-shot session task for 2 epochs. The training optimizer is adopted as Adam (Kingma, 2014) with a learning rate of 0.0005 throughout all the training phase for all datasets.

### 5.2 MAIN RESULTS

In this section, we report the comparison results of our proposed method with the previously mentioned baselines. The detailed accuracy for each session, the average accuracy and the performance drop on the CIFAR100, CUB200-2011 and miniImageNet datasets are shown in Table 1, Table 2 and Table 3, respectively.

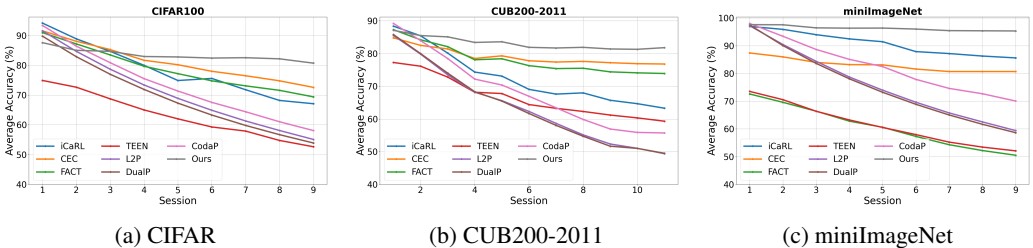

(a) CIFAR        (b) CUB200-2011        (c) miniImageNet

Figure 3: Illustration of Top-1 accuracy in each session on three datasets. Our proposed method outperforms all comparison methods on all datasets.

Table 1: Performance comparison on the CIFAR-100 dataset across three evaluation metrics: Top-1 accuracy $A_t$ for each task, average accuracy $A_{avg}$, and performance dropping ($PD$). $\uparrow$ indicates higher is better and $\downarrow$ indicates lower is better. $\dagger$ indicates the results reported from (Liu et al., 2024). $\ddagger$ indicates the results reproduced by ourselves.

| Methods | Top-1 accuracy on CIFAR100 (%) | | | | | | | | | $A_{avg}\uparrow$ | $PD\downarrow$ |
|---|---|---|---|---|---|---|---|---|---|---|---|
| | 0 | 1 | 2 | 3 | 4 | 5 | 6 | 7 | 8 | | |
| *Conventional CL & FSCIL* | | | | | | | | | | | |
| iCaRL[†] (Rebuffi et al., 2017) | 94.2 | 88.9 | 84.7 | 80.0 | 74.9 | 75.6 | 71.8 | 68.2 | 67.1 | 78.4 | 27.1 |
| CEC[†] (Zhang et al., 2021) | 91.6 | 88.1 | 85.3 | 81.7 | 80.2 | 78.0 | 76.5 | 74.8 | 72.6 | 81.1 | 19.0 |
| FACT[†] (Zhou et al., 2022) | 91.0 | 87.2 | 83.5 | 79.7 | 77.2 | 74.8 | 73.1 | 71.6 | 69.4 | 78.6 | 21.6 |
| TEEN (Wang et al., 2024) | 74.9 | 72.6 | 68.7 | 65.0 | 62.0 | 59.3 | 57.9 | 54.8 | 52.6 | 63.1 | 22.28 |
| *CL based on FD models* | | | | | | | | | | | |
| L2P[‡] (Wang et al., 2022b) | 91.6 | 84.7 | 78.7 | 73.4 | 68.9 | 64.9 | 61.3 | 58.1 | 55.1 | 70.7 | 36.5 |
| DualP[‡] (Wang et al., 2022a) | 89.8 | 82.9 | 77.0 | 71.9 | 67.3 | 63.3 | 59.8 | 56.7 | 53.9 | 69.2 | 35.9 |
| CodaP[‡] (Smith et al., 2023) | 93.4 | 86.4 | 80.9 | 75.5 | 71.4 | 67.6 | 64.4 | 61.0 | 58.1 | 73.2 | 35.3 |
| **SVFCL (Ours)** | 87.6 | 85.0 | 84.7 | 83.0 | 82.8 | 82.4 | 82.6 | 82.2 | 80.7 | **83.5** | **6.9** |

Qualitatively, we illustrate the Top-1 comparison curve for all three datasets in Fig. 3, where our method SVFCL (represented by the gray line), outperforms all other comparison methods by a significant margin and exhibits a smaller performance drop.

Quantificationally, as shown in Table 1, Table 2 and Table 3, SVFCL achieves the highest Top-1 accuracy of 83.5%, 83.5%, and 96.3%, along with the lowest performance drops of 6.9, 4.5, and 2.3 on the three datasets, respectively. Specifically, SVFCL surpasses the second best CEC by 2.4% and the best prompt-based CL method CEC by 10.3% on CIFAR100. On the CUB200-2011 dataset, SVFCL achieves significant improvements of 4.4% over the best conventional method CEC, and 15.1% over the best prompt-based CL method CodaP. In the evaluations on miniImageNet, there is a significant margin of 5.4% in Top-1 accuracy between SVFCL and the second-best method iCaRL. These results demonstrate the robust capacity of our method in learning incremental tasks and mitigating catastrophic forgetting. Besides, we find that prompt-based CL methods sometimes fail to outperform conventional FSCIL methods such as CEC and FACT, which may be due to their continual updating during incremental sessions. In contrast, our method fine-tunes the model across all sessions, while it performs well with no obvious forgetting.

## 5.3 DISCUSSION

To gain a deeper understanding of the effectiveness of the proposed SVFCL model, we conduct an extensive ablation study to examine which blocks should be fine-tuned in ViT for FSCIL and to what extent low-rank approximation in SVD should be applied for further reducing the learnable in SVFCL. Specifically, we answer the following questions: **(1) Where to employ SVFCL within ViT? (2) Can learnable parameters be further reduced in SVFCL?** Additionally, we compare our method with three most recent FSCIL methods based on foundation models and answer the following question: **(3) How does the performance compared with methods based on foundation models?**

Table 2: Performance comparison on the CUB200-2011 dataset across three evaluation metrics: Top-1 accuracy $A_t$ for each task, average accuracy $A_{avg}$, and performance dropping ($PD$). ↑ indicates higher is better and ↓ indicates lower is better. [†] indicates the results reported from (Liu et al., 2024). [‡] indicates the results reproduced by ourselves.

| Methods | Top-1 accuracy on CUB200-2011 (%) | | | | | | | | | | | $A_{avg}\uparrow$ | $PD\downarrow$ |
| | 0 | 1 | 2 | 3 | 4 | 5 | 6 | 7 | 8 | 9 | 10 | | |
|---|---|---|---|---|---|---|---|---|---|---|---|---|---|
| *Conventional CL & FSCIL* | | | | | | | | | | | | | |
| iCaRL[‡] (Rebuffi et al., 2017) | 88.4 | 85.4 | 80.1 | 74.3 | 73.1 | 69.0 | 67.6 | 68.0 | 65.7 | 64.7 | 63.3 | 72.7 | 25.1 |
| CEC[†] (Zhang et al., 2021) | 84.8 | 82.5 | 81.4 | 78.5 | 79.3 | 77.8 | 77.4 | 77.6 | 77.2 | 76.9 | 76.8 | 79.1 | 8.0 |
| FACT[†] (Zhou et al., 2022) | 87.3 | 84.2 | 82.1 | 78.1 | 78.4 | 76.3 | 75.4 | 75.5 | 74.4 | 74.1 | 73.9 | 78.1 | 13.4 |
| TEEN (Wang et al., 2024) | 77.3 | 76.1 | 72.8 | 68.2 | 67.8 | 64.4 | 63.2 | 62.3 | 61.2 | 60.3 | 59.3 | 66.6 | 17.95 |
| *CL based on FD models* | | | | | | | | | | | | | |
| L2P[‡] (Wang et al., 2022b) | 85.5 | 79.8 | 73.5 | 68.2 | 65.6 | 62.3 | 58.6 | 55.1 | 52.3 | 51.0 | 49.4 | 63.8 | 36.1 |
| DualP[‡] (Wang et al., 2022a) | 85.8 | 80.0 | 74.0 | 68.3 | 65.5 | 61.7 | 58.0 | 54.7 | 51.7 | 51.0 | 49.5 | 63.6 | 36.3 |
| CodaP[‡] (Smith et al., 2023) | 89.2 | 84.0 | 78.5 | 72.2 | 70.4 | 66.9 | 63.4 | 59.8 | 56.9 | 55.9 | 55.7 | 68.4 | 33.5 |
| **SVFCL (Ours)** | 87.1 | 85.4 | 85.0 | 83.6 | 83.7 | 82.0 | 81.9 | 82.3 | 82.2 | 82.2 | 82.6 | **83.5** | **4.5** |

Table 3: Performance comparison on the minImageNet dataset across three evaluation metrics: Top-1 accuracy $A_t$ for each task, average accuracy $A_{avg}$, and performance dropping ($PD$). ↑ indicates higher is better and ↓ indicates lower is better. [†] indicates the results reported from (Park et al., 2024; D'Alessandro et al., 2023). [‡] indicates the results reproduced by ourselves.

| Methods | Top-1 accuracy on miniImageNet (%) | | | | | | | | | $A_{avg}\uparrow$ | $PD\downarrow$ |
| | 0 | 1 | 2 | 3 | 4 | 5 | 6 | 7 | 8 | | |
|---|---|---|---|---|---|---|---|---|---|---|---|
| *Conventional CL & FSCIL* | | | | | | | | | | | |
| iCaRL[‡] (Rebuffi et al., 2017) | 97.0 | 95.9 | 94.0 | 92.5 | 91.5 | 87.9 | 87.2 | 86.3 | 85.6 | 90.9 | 11.4 |
| CEC[†] (Zhang et al., 2021) | 87.4 | 86.0 | 84.0 | 83.2 | 83.1 | 81.6 | 80.7 | 80.7 | 80.7 | 83.0 | 6.7 |
| FACT[†] (Zhou et al., 2022) | 72.6 | 69.6 | 66.4 | 62.8 | 60.6 | 57.3 | 54.3 | 52.2 | 50.5 | 60.7 | 22.1 |
| TEEN (Wang et al., 2024) | 73.5 | 70.5 | 66.4 | 63.2 | 60.5 | 57.9 | 55.2 | 53.4 | 52.1 | 61.4 | 21.4 |
| *CL based on FD models* | | | | | | | | | | | |
| L2P[‡] (Wang et al., 2022b) | 97.2 | 90.4 | 84.4 | 78.7 | 74.0 | 69.6 | 65.8 | 62.5 | 59.4 | 75.8 | 37.8 |
| DualP[‡] (Wang et al., 2022a) | 97.5 | 90.1 | 83.6 | 78.1 | 73.2 | 68.9 | 65.0 | 61.7 | 58.6 | 75.2 | 38.9 |
| CodaP[‡] (Smith et al., 2023) | 98.0 | 93.4 | 88.7 | 85.1 | 82.5 | 77.9 | 74.6 | 72.7 | 70.1 | 82.5 | 28.0 |
| **SVFCL (Ours)** | 97.6 | 97.6 | 96.5 | 96.3 | 96.4 | 96.0 | 95.5 | 95.4 | 95.3 | **96.3** | **2.3** |

**Where to employ SVFCL within ViT?** The proposed SVFCL module can be integrated into any block of the pre-trained ViT. Intuitively, applying SVFCL to more blocks may enhance plasticity for new classes but also increase the number of learnable parameters, raising the risk of overfitting. To strike a balance between efficiency and performance, we conduct an ablation study on the CUB200-2011 dataset, progressively applying SVFCL to an increasing number of blocks. The results are illustrated in Fig. 5, indicating that using SVFCL in blocks 0-6 yields

| Blocks | Average Accuracy (%) |
|---|---|
| 0-6 | 83.46 |
| 3-8 | 83.12 |
| 9-11 | 82.80 |

Table 4: Comparison results of employing SVFCL in 6 blocks located at different positions in ViT on CUB200-2011.

optimal performance. Additionally, we compare the impact of applying SVFCL to the same number of blocks at different positions in ViT, as shown in Table 4. The results reveal that tuning the first 6 blocks significantly outperforms tuning the last 6 blocks. Based on these findings, we insert SVF into blocks 0-6 of the pre-trained ViT across all our experiments.

**Can learnable parameters be further reduced in SVFCL?** In the proposed SVFCL, we utilize full SVD to re-parameterize the adapter weights of the two MLPs in each block. However, the full SVD requires calculating and tuning all singular values for each few-shot session, which may be redundant because the weights of MLP in each block are typically low rank, as validated in Fig. 7a of the Appendix B. Consequently, we replace the full SVD with a low-rank approximation SVD in SVFCL, reducing the number of singular values and thereby further decreasing the learnable

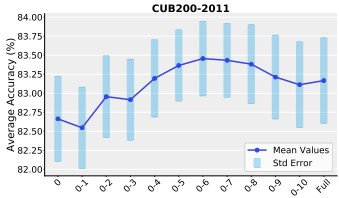

Figure 4: Ablation study of the low-rank approximation SVD in SVFCL on the datasets CUB200-2011 and miniImageNet with varying ranks.

Figure 5: Ablation study of fine-tuning different blocks within ViT.

Table 5: Performance comparison with FSCIL methods based on foundation models. We compare with three most recent state-of-the-art methods: CPE-CLIP (D'Alessandro et al., 2023), PriVi-Lege (Park et al., 2024), and ASP (Liu et al., 2024). ‡ indicates the results reproduced by ourselves for ASP on the miniImageNet dataset.

| Methods | CUB200-2011 (%) | | | | $A_{avg} \uparrow$ | $PD \downarrow$ | miniImageNet (%) | | | | $A_{avg} \uparrow$ | $PD \downarrow$ |
|---|---|---|---|---|---|---|---|---|---|---|---|---|
| | 0 | 3 | 6 | 10 | | | 0 | 3 | 5 | 8 | | |
| CPE-CLIP | 81.6 | 71.9 | 67.7 | 64.6 | 70.8 | 17.0 | 90.2 | 86.8 | 85.1 | 82.8 | 86.1 | 7.4 |
| PriViLege | 82.2 | 77.8 | 75.7 | 75.1 | 77.5 | 7.1 | 96.7 | 95.5 | 94.9 | 94.1 | 95.3 | 2.6 |
| ASP‡ | 87.1 | 83.4 | 82.6 | 83.5 | **83.8** | **3.6** | 93.3 | 90.6 | 89.3 | 88.4 | 90.3 | 4.9 |
| **SVFCL (Ours)** | 87.1 | 83.6 | 81.9 | 82.6 | 83.5 | 4.5 | 97.6 | 96.3 | 96.0 | 95.3 | **96.3** | **2.3** |

parameters. Specifically, we choose $r' < r$ to re-parameterize the weight matrix $M$ as $\sum_{i=1}^{r'} s_i \mathbf{u}_i \mathbf{v}_i^\top$, in which we can only tune the singular values $s_i$ with $i = \{1, \cdots, r'\}$. To demonstrate the feasibility of the low-rank approximation SVD in SVFCL, we conduct experiments on the two datasets, as shown in Fig. 4. The results show that the performance of low-rank approximation SVD significantly impacts the performance of SVFCL, and excessively coarse approximations, such as setting the rank to 50 or 100, leading to performance degradation on both datasets. Within an acceptable range, we can set the rank $r'$ as 500 to reduce model parameters and efficient fine-tuning without significantly compromising the performance our method.

**How does the performance compared with methods based on foundation models?** To further validate the effectiveness of our method, we compare our method with three newly proposed FSCIL methods: CPE-CLIP (D'Alessandro et al., 2023), PriViLege (Park et al., 2024), ASP (Liu et al., 2024) as shown in Table 5. We consider these methods separately because they primarily utilize the pre-trained models and employ prompt-tuning techniques, providing more discriminative features in few-shot training phases. Note that even though CPE-CLIP and PriViLege further leveraged additional textual information to enhance the FSCIL performance based on CLIP (Radford et al., 2021), our proposed SVFCL successfully becomes new state-of-the-art method on the FSCIL setting. Specifically, SVFCL outperforms CPE-CLIP by 12.7% on CUB200-2011 and 10.2% on miniImageNet, as well as surpassing PriViLege by 6.0% on CUB200-2011 and 1.0% on miniImageNet.

## 6 CONCLUSION

In this work, we focus on jointly mitigating both catastrophic forgetting and overfitting in the FSCIL scenario, particularly when leveraging pre-trained models. We propose a simple yet effective framework that presents the SVFCL strategy to fine-tune the pre-trained ViT backbone for each incremental session. SVFCL effectively addresses the challenges of overfitting and catastrophic forgetting while incurring relatively few learnable parameters and minimal computational overhead. Our final experiments demonstrate that SVFCL achieves promising performance improvements compared to various state-of-the-art methods. In the future, we aim to further explore the feasibility of integrating our SVFCL with other advanced architectures to enhance its performance and applicability across diverse datasets.

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

## A   DETAILED VISUALIZATION

We further compare the visualization of incremental gradient focus with two prompt-tuning based methods: L2P (Wang et al., 2022b) and DualP (Wang et al., 2022a). As illustrated in Figure 6, full-tuning method struggles to learn effective features and tends to overfit to local areas which are forgotten in subsequent sessions. Our method learns critical features in the beginning, while L2P and DualP diffuse their attention on background regions.

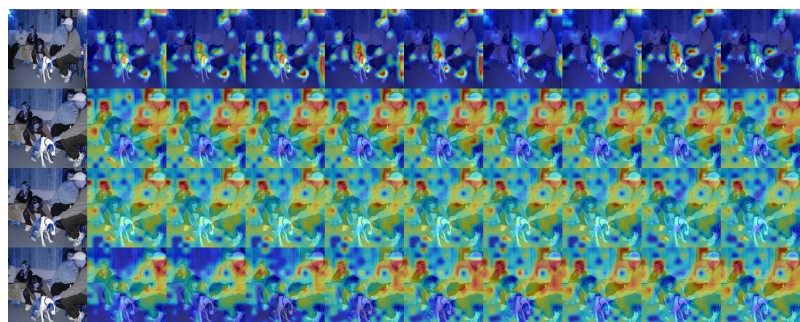

Figure 6: The visualization of incremental gradient focus compared with (Top) Full-tuning, (Top second) L2P, (Top third) DualP and (Bottom) Ours on miniImageNet. Our method focuses attention on crucial regions while Both L2P and DualP are affected by background regions to varying degrees and the Full-tuning method struggles to learn effective features.

## B   ANALYSIS OF THE SINGULAR VALUES

To examine the singular values of different layers, we visualize them across various blocks, as shown in Fig. 7. It is evident that the large singular values are limited, which forms the foundation for our proposed SVFCL method.

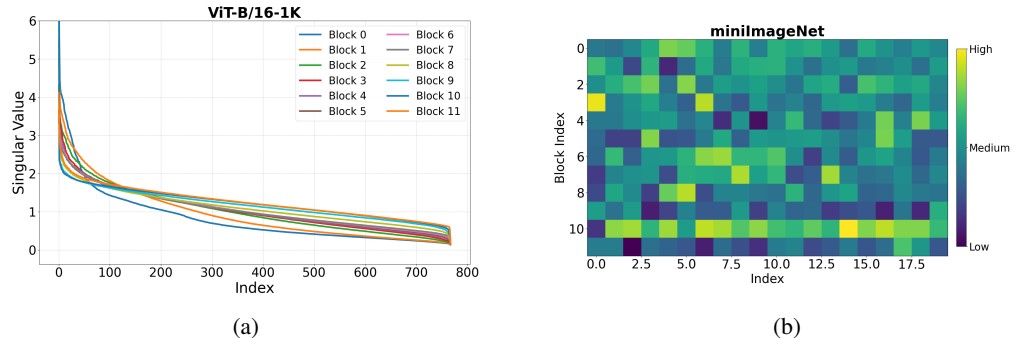

(a)                                             (b)

Figure 7: (a) The singular values of the pre-trained weights of the first fully-connected layers within MLP modules. The singular values exhibit a long-tailed distribution, with the majority of singular values being relatively small. (b) The corresponding singular values (showing only the top 30 indices) after training on the base task, most of which are also relatively small, reflecting the task's bias towards the bases.

