# OpenReview forum: "Singular Value Fine-tuning for Few-Shot Class-Incremental Learning"
_ICLR.cc/2025/Conference — ICLR 2025 Conference Withdrawn Submission_

### Official Review · Reviewer_drNJ · 2024-11-02

**Soundness:** 2
**Presentation:** 2
**Contribution:** 2
**Rating:** 3
**Confidence:** 4

**Summary:**

This paper proposes a Singular Value Fine-tuning for few-shot Class-incremental Learning (SVFCL). SVFCL incorporates incremental adapters based on the pre-trained ViT encoder, which tries to reduce the risk of overfitting.

**Strengths:**

Using singular value decomposition is funny.

**Weaknesses:**

I think this paper cannot capture the author's core idea, and there are many contradictions in this paper:

From the method:
1) In the abstract, the authors say that SVFCL contains only a small number of learnable parameters, but in Section 4, SVFCL fine-tunes the backbone of the foundation model. The authors use ViT-B/16-1K as the backbone, which has a large number of parameters.
2) I think the biggest challenge with fine-tuning the backbone is catastrophic forgetting, and fine-tuning the matrix S can retain the previous knowledge and adapt to the new task. However, the authors focus on overfitting and mention that the solution to overfitting is to reduce the number of parameters for fine-tuning in Section 4. Compared with SVFCL, other methods freeze the backbone and only fine-tune the classifier, which fine-tune fewer parameters and solve overfitting better.
3) I think the authors are vague about the description of the adapters. Where are adapters used? And whether there are additional parameters? In the contributions, adapters introduce quite a few learnable parameters, are they MLP? But in Section 4, the authors finetune the matrix S. If additional parameters are available, introducing the number of parameters will still exacerbate the overfitting.
4) The authors propose to fine-tune the backbone. Whether the entire backbone do the singular value fine-tuning, which undoubtedly needs to adjust a huge number of parameters?
5) The authors update the matrix S in the method. Why only update S? What is the effect if update U or V? what is the effect if update two matrices?
6) In the experiment, the authors mention the use of MLP. So will the MLP perform singular value decomposition? Will it fine-tune in each session? In addition, the authors add an MLP after 12 ViT blocks, which undoubtedly increases the number of parameters, will it further lead to overfitting?

From the writing:
1) The method focuses on FSCIL, but the authors spend a lot of space describing CIL in the abstract.  In addition, the authors propose the use of adapters in the abstract and the contributions, but do not introduce this module in detail in Section 4.
2) Figure 1 does not indicate which dataset the example comes from, is it from the base session or the incremental sessions? And I think the author should include more examples in Appendix A instead of just a few more visualizations of other methods.
3) The expression of Figure 2 is very vague. The caption shows this figure represents the whole framework, but it only describes the weights for parts of the model. Meanwhile, the authors freeze the pre-trained parameters. What does the red part indicate? And where are all MLPs used by the author?
4) I think the first contribution is the same as the second contribution. The first contribution is only the application of the adapter, which appears limited and may not be entirely convincing.
5) The authors also have some inconsistent descriptions and grammatical errors, such as the authors describe SVFCL as a strategy in the contribution but a framework in Figure 2.

From the experiment:
1) The role of Figure 3 and the three tables is consistent, why do the authors repeat the experimental results?
2) I think it would be fairer for the authors to compare some FSCIL or CIL methods that use ViT or Transformer, such as [1-4].

[1] Pre-trained Vision and Language Transformers Are Few-Shot Incremental Learners. CVPR 2024.

[2] Calibrating Higher-Order Statistics for Few-Shot Class-Incremental Learning with Pre-trained Vision Transformers. CVPR 2024.

[3] Semantic-visual Guided Transformer for Few-shot Class-incremental Learning. ICME 2023.

[4] Ranpac: Random projections and pre-trained models for continual learning. NIPS 2023.

3) The authors use only 2 epochs in each incremental session, is the model underfitting? The authors use 3 epochs in the base session, whether the high accuracy relies on knowledge leakage from pre-trained models.
4) I think the authors should add the analysis of matrix U, S, and V in the Discussion, and what role U and V play in the model.

**Questions:**

See the Weaknesses.

---

### Official Review · Reviewer_fNqT · 2024-11-02

**Soundness:** 3
**Presentation:** 3
**Contribution:** 2
**Rating:** 5
**Confidence:** 4

**Summary:**

This paper proposed a Singular Value Fine-tuning approach to constantly learn base and incremental sessions based on the pre-trained ViT encoder. The proposed approach achieved impressive results on several benchmarks.

**Strengths:**

1. The proposed Singular Value Fine-tuning approach is reasonable and technically feasible.
2. The experiments results show that the proposed approach outperforms existing methods.
3. The paper is well organized and easy to follow.

**Weaknesses:**

1. What are the key differences between the proposed Singular Value Fine-tuning and LoRA (Low-Rank Adaptation)? LoRA is widely used for fine-tuning large models by adjusting a small number of parameters through matrix decomposition to adapt to new tasks, which is similar to the motivation and techniques of this paper.
2. Line 305-307: "We refrain from updating the singular matrices U and V , indicating that we maintain the same singular vectors (semantic clues) across all incremental sessions". How the singular vectors capture the semantic clues of newly added classes under FSCIL settings, the authors should provide more explanation or evidence to support this claim.
3. Concerns about data leakage: When pre-trained large models undergo incremental learning, there is a high likelihood that the new class data will overlap with the pre-training data, which contradicts the assumptions of FSCIL. I would like to hear the authors' views on this issue.

**Questions:**

Please kindly refer to the weaknesses.

---

### Official Review · Reviewer_2ZpS · 2024-11-03

**Soundness:** 3
**Presentation:** 3
**Contribution:** 2
**Rating:** 5
**Confidence:** 4

**Summary:**

The paper introduces Singular Value Fine-Tuning for Class-Incremental Learning (SVFCL), a method to tackle forgetting and overfitting in Few-Shot Class-Incremental Learning (FSCIL). Using Vision Transformer (ViT), it fine-tunes only key singular values for each task, reducing parameters to prevent overfitting. SVFCL outperforms current methods while maintaining efficiency.

**Strengths:**

Pros:
1.	The method achieves state-of-the-art (SOTA) results on several FSCIL benchmarks.
2.	The paper is well-written and structured, making it easy to follow.

**Weaknesses:**

Cons & Questions:
1.	The approach of updating the singular value matrix of each network parameter closely resembles null-space based methods, which are already well-studied and widely applied in class-incremental learning tasks. Please clarify how this approach differs from these existing methods and explain the novelty of the overall idea, as this similarity raises concerns about the originality of the proposed framework.
[a] Saha, Gobinda, Isha Garg, and Kaushik Roy. "Gradient projection memory for continual learning." ICLR 2021.
[b] Kong, Yajing, et al. "Balancing stability and plasticity through advanced null space in continual learning." ECCV 2022.
[c] Zhao, Zhen, et al. "Rethinking gradient projection continual learning: Stability/plasticity feature space decoupling." CVPR 2023.
2.	While the proposed framework only requires learning a few adapters, it still necessitates updating all network parameters per Eq. (6) by processing each input x of all classes, which does not seem to align with claims of being lightweight.
3.	The proposed method performs worst on the base classification tasks across all datasets when using the same ViT-based backbone. Please explain why.
4.	There are some typos in the paper, such as “rataining” in Line 012 and “minImageNet” in Line 448.

**Questions:**

please refer to **weakness**.

**Details Of Ethics Concerns:**

N.A.

---

### Note · Authors · 2024-11-14

I have read and agree with the venue's withdrawal policy on behalf of myself and my co-authors.